# Mountain Meadows and Glades of the Carpathians—Type or Element of Landscape? The Problem of Delimitation and Typology of Mountain Pasture Landscapes

**Michał Sobala**

Faculty of Natural Sciences, University of Silesia, Będzińska 60, 41200 Sosnowiec, Poland; michal.sobala@us.edu.pl; Tel.: +48-3236-89-263

**Abstract:** The typologies of landscapes of individual states that have ratified the European Landscape Convention do not include mountain pasture landscapes. Pasture landscapes in the Carpathians are preserved in a relatively good condition, although their spatial extent has considerably shrunk over the last few decades. The article demonstrates that mountain meadows and glades in the Carpathians meet all the conditions that allow them to be classified as a type of landscape, and thus they should be included in national typologies of landscapes. Firstly, they constitute a set of natural (non-forest vegetation) and anthropogenic (traditional shepherding buildings) objects. Secondly, they are a dynamic system in which natural, social and economic processes take place. They are also a source of stimuli, affecting different human senses and values and are a system that provides various real and potential services. The inclusion of mountain pasture landscapes in national typologies may provide a stronger basis for their protection.

**Keywords:** national typologies of landscape; pasture landscape; landscape conservation; mountain meadows; glades; the Carpathians; the Beskids

---

## 1. Introduction

Pasture landscapes are mostly located in agriculturally marginalised areas where commercial agriculture is constrained due to harsh natural conditions such as subtropical dry lands, temperate mountain uplands or sandy heathlands [1]. They are a subtype of cultural landscape, the function of which is subordinate to extensive animal husbandry [2]. In the course of the development of agricultural civilizations, various systems of animal husbandry and corresponding types of pasture landscapes have developed; for instance, nomadism, transhumance, summer mountain grazing, grazing of cattle, sheep, horses and pigs in forests, or reindeer breeding [3–5]. These different subtypes reflect the local specificity of the landscape zones [6,7].

Because of their marginalised location, pasture landscapes have usually not undergone the processes of intensification, enclosure and specialisation characteristic of urbanised core agricultural areas. Unfortunately, the economic marginalisation of these pastoral areas means that they have become vulnerable to depopulation and abandonment due to poor social and physical infrastructure, and have become co-dependent on supplementary economic activities [8,9].

The importance of pasture landscapes relates to their role in maintaining biodiversity [10] and for providing livelihood from livestock husbandry [11]. As a result of pastureland abandonment, the rich and diverse ecological and cultural heritage of pasture landscapes created over the centuries is lost. However, the abandonment of pasturelands and their rewilding suits the interests of some groups, such as the wilderness recreation industry, rewilding enthusiasts, economic planners and

agro-environmental managers [12,13]. What is more, in conventional views, pastoralism was classified as a stage of civilization that needed to be abolished and transcended in order to reach a higher level of development. In Central Asia, the 20th century experienced a variety of concepts to sedentarize nomads and to modernize their lifestyle. Modernization theory captured all walks of life and tried to optimize breeding techniques, pasture utilization, transport and processing concepts. Nevertheless, pastoralism should be perceived as a flexible strategy to adapt the use of marginal resources in remote locations with difficult access, rather than a transitory stage on the path to modern development [14].

Unfortunately, pasture landscapes nowadays have a relict character [15,16]. The changes in pasture landscapes extent following abandonment have been documented mainly for Europe [17–22], South America [23] and Asia [1,24]. The structure and functions of traditional pasture landscapes have survived in a relatively unchanged form only in Central Asia [24]. Currently, pasture landscapes coexist with other types of landscapes that have developed independently or have arisen as a result of a change in the lifestyle of the communities that previously lived by grazing animals. These negative changes have created new challenges for the management of grasslands [12,25]. The search for alternative futures remains urgent and is a challenge for landscape research [26]. There is a need to introduce active conservation of the landscape, consisting mainly of selective tree logging and conservation of valuable non-forest communities as well as the implementation of programmes aimed at maintaining or even revitalizing local pastoral traditions affecting the local economy and landscape [5]. In highly developed countries today, the maintenance of pasture landscape depends on the efforts of biodiversity conservation [9].

In parallel with grassroots movements, there are some top-down initiatives. The European Landscape Convention was adopted by the Committee of Ministers of the Council of Europe on 19 July 2000 in Strasbourg and opened for signature of the Member States of the Organisation in Florence on 20 October 2000. The Convention came into force on 1 March 2004 and 39 Council of Europe member states have ratified the Convention so far. It aims to promote European landscape protection, management and planning, and to organise European co-operation. The ratification of the European Landscape Convention resulted in a series of actions aimed at establishing and implementing landscape policy. One of the means of implementing the Convention is the identification and assessment of the state of preservation of landscapes in each country [27]. As a result of this, many European countries have begun their work on creating completely new landscape typologies or modifying old ones, taking into account the message and objectives of the Convention [28–37]. Different initiatives have also been developed on a pan-European level to identify and classify landscapes in Europe. The maps of Milanova and Kushlin [38] and Meeus [39] were a first attempt to produce a European Landscape Map. However, they were rather inaccurate due to a lack of systematic digital information with a high-spatial accuracy and computer-supported data processing. Subsequently, the Pan-European Landscape Map (LANMAP) has been produced, based on digital data sets with a high-spatial accuracy and a high degree of flexibility to enable adaptations and extensions [40]. Nevertheless, important management dimensions, including land management intensity, have not been included in these initiatives. To fill this gap, a Europe-wide spatially-explicit typology and inventory of agricultural landscapes was developed. The Europe-wide datasets representing land cover, land management intensity and landscape structure on a 1 km$^2$ resolution were used. It can be seen as a first step towards a comprehensive regional framework for comparison of agricultural landscapes across Europe [41]. Both Pan-European and national typologies in most cases include entire national territories and usually take into account the hierarchical structure of landscape units [42]. For example, LANMAP distinguishes 350 European landscape types at four levels, whereas a typology of current Polish landscapes distinguishes 3 groups, 15 types and 49 subtypes of landscapes [43]. The last one has been tested in several regions of Poland [44–47]. These tests were aimed at determining the suitability of the proposed typology for practical activities in the field of landscaping, including the preparation of landscape audits throughout the country [48]. The need to test the typology on at least

a few physical-geographical, cultural or socio-economic regions was indicated by its authors, stressing that the conclusions drawn should be used to improve the presented proposal.

In many national typologies of landscapes, the author did not distinguish a separate type of mountain pasture landscape. For instance, in Poland, only the landscapes of mountain meadows above the forest border are distinguished. Nevertheless, there are still vast areas of mountain meadows and glades within the forest montane zone in the Carpathians formed as a result of traditional pastoral activity. Similarly, in the Slovak typology, this type of landscape was not distinguished [49]. Identifying the pasture landscapes as a separate type or subtype could prove very important for the effectiveness of landscape conservation and management [50], especially in the context of landscape audit instructions [51]. According to the typology of Poland's current landscapes, forest landscapes prevail in the Polish Carpathians, and the mid-forest meadows and glades located within them as structural elements within the forest background can be considered landmarks [43]. A similar situation relates to the Slovak Carpathians [49] and Alps in Slovenia [52].

The author, with this in mind, aimed to investigate whether the proposed national typologies of landscapes should include mountain pasture landscapes. It is essential, as top-down initiatives may have a great influence on pasture landscape protection. It must be emphasised that public policies often face difficulties in that area [22,53]. Research on the changes of forest ecosystems and pasture landscapes conducted in the Western Carpathians induced the author to undertake this task.

According to the European Landscape Convention, landscape is understood as an area perceived by people whose character is the result of the action and interaction of natural and/or human factors. Landscape in a broad sense may be treated as [43]:

- A set of material objects with a specific content (i.e., physical, chemical and biological composition) and form (landform and texture, and in the case of anthropogenic elements—also with a specific composition).
- A system of related processes integrating material objects.
- A set of stimuli that affect the various senses of the user, particularly a set of views and panoramas with specific aesthetic value.
- A set of natural, social, economic, material, spiritual, historical, physiognomic, aesthetic, symbolic and other values (potentials).
- A system that provides real and potential services (benefits) for different groups of users.

It should be emphasised that only the first two categories are of objective nature, existing independently of the will, views and attitudes of the user. The other categories are of subjective nature, connected to social perception.

In turn, to the separation of landscape units Solon et al. [51] proposed the following criteria:

- Homogeneity of the landscape background while maintaining spatial heterogeneity.
- Preservation of functional connections between spatial elements of the landscape.
- Repeatability of spatial structure and physiognomy in different parts of the landscape (this condition may not always be preserved, especially in the case of unique landscapes).

Within the boundaries of landscape units, there may be objects and complexes of objects, as well as patches of land cover, with particular characteristic (or typically, classically developed) features, which can be referred to as landmarks [54,55]. According to the existing landscape typologies, mountain meadows and glades are considered landmarks within forest landscapes.

Respecting the scope of the definition of landscape, this article will demonstrate, using the selected example of the Beskids, that the Carpathians mountain meadows and glades can be treated as a type/subtype of landscape. The Beskids glades will be described in terms of five aspects of the landscape that constitute its holistic concept.

## 2. Materials and Methods

### 2.1. Study Area

The Beskids (Western Carpathians) stretches for about 600 km from the Cheremosh River in the east to the Bečva River in the west, and reaches a width of about 50–70 km. Detailed research was conducted within 3 glades situated in the Western Beskids (Figure 1).

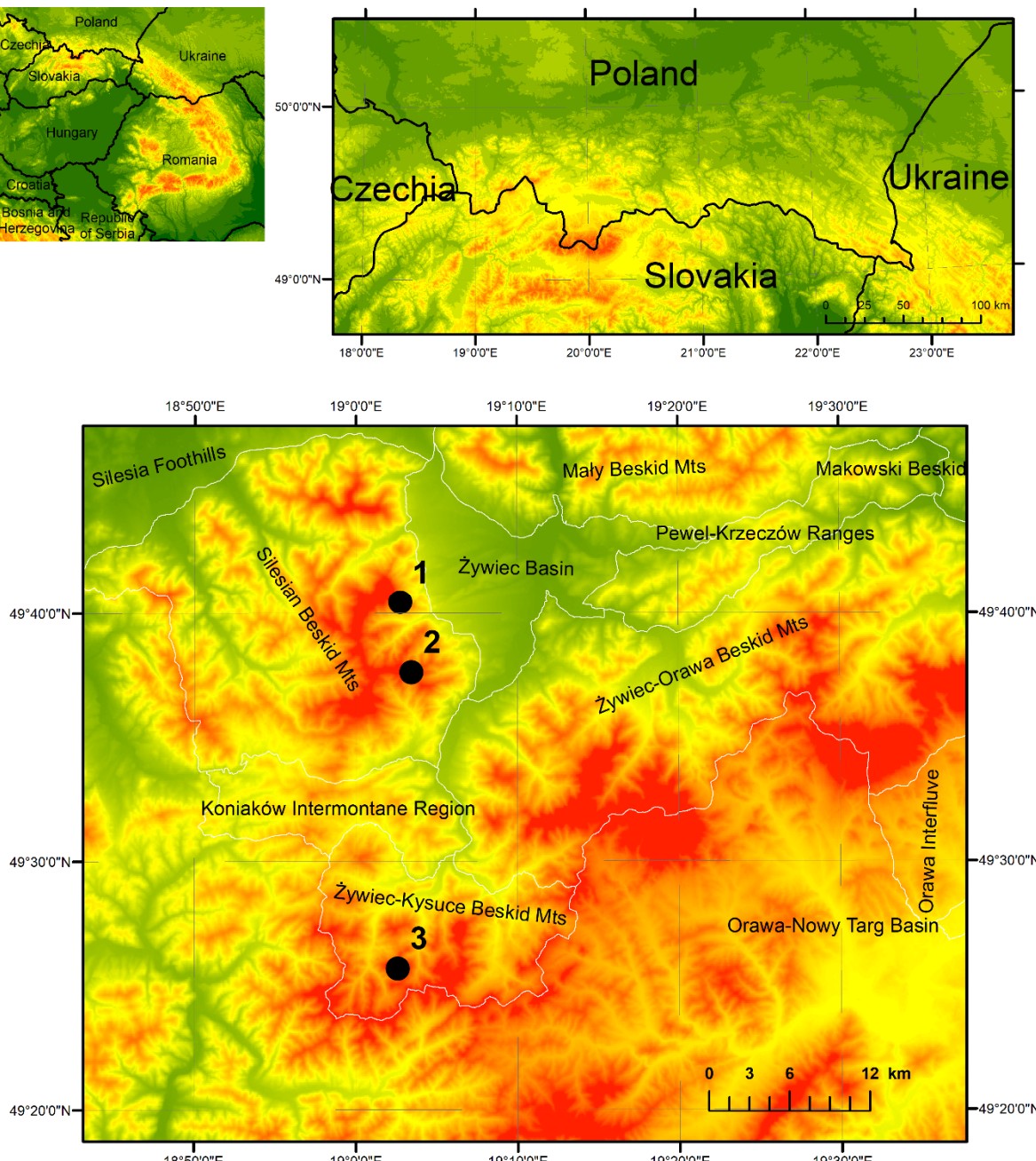

**Figure 1.** Location of the study area: 1—Jaskowa glade, 2—Radziechowska glade, 3—Bendoszka glade.

The Beskids are characterized by medium and low mountain relief with steep slopes (mean elevation > 800 m a.s.l.). The areas span over three vertical climatic zones, namely, moderate warm (with mean annual temperature >6 °C), moderate cool (4–6 °C) and cool (<4 °C). Precipitation on the highest ridges reaches 1300 mm year$^{-1}$. These conditions of the area are suitable to forests and grasslands, whereas current land use reflects the socio-political and economic conditions in the late

19th century, when the Beskids region was heavily populated and large areas were deforested for agricultural use, despite unfavourable topographic, climatic and soil conditions [56]. Numerous clearings with farmsteads are located in flat areas of the forest, and mountain ridges include pasture grounds which recently decreased in area due to the abandonment of breeding farm management [57].

Pasture landscapes in the Carpathians have been preserved in a relatively good condition, although their spatial extent has shrunk considerably over the last few decades [56,58–61]. The process of abandoning the traditional pastoral economy, which has been underway for decades, has also contributed to the disappearance of its constituent elements, including, in particular, traditional wooden buildings (huts and sheds) [57].

## 2.2. Analysis of Objective Nature of Landscape

The analysis of objective nature of landscape consists of evaluation of state and changes in landscape over time. To this end, only maps with similar purposes based on direct field mapping were selected. However, serious difficulties to match such criteria were encountered. Therefore, only six of the seven maps acquired were used in this study (Table 1).

**Table 1.** Cartographic materials used for the analyses.

| Map Type | Year | Scale Resolution |
|---|---|---|
| Austrian cadastral maps | 1848 | 1:2880 |
| Spezialkarte der Österreichisch-Ungarischen Monarchie | 1879–1885 | 1:75,000 |
| WIG military map | 1933 | 1:100,000 |
| Military topographic map | 1960–1975 | 1:25,000 |
| Topographic map of Poland | 1979 | 1:10,000 |
| Orthophotomap | 2015 | $0.25 \times 0.25$ m |

All the historical maps were georeferenced in two steps, which consisted in calculating the transformation matrix, carrying out proper geometric transformation and interpolation resampling of a distorted image to a new raster of regular size (i.e., the so-called "rubbersheeting"). Such a two-step process allowed a higher georeferencing accuracy, which ensures the quality of results obtained and increases the confidence in the conclusions. In each case, georeferencing was specifically adjusted to the quality and type of data, to achieve the best possible results for each series. Hence, Austrian cadastral maps were overlaid onto a grid with a size corresponding to the map frame size using affine transformation and the coordinates of the frame corners [62]. Rectification was then carried out and its precision verified by estimating the root-mean-square error (RMSE), which was < 4.91 m for each map sheet. The Spezialkarte der Österreichisch-Ungarischen Monarchie was georeferenced only by means of control points of the reference layer using the affine transformation. This kind of georeferencing of a single map sheet gives better results than that based on fitting the corners in the millimeter mesh grid [63]. The military maps were georeferenced by overlaying the corner points of the raster image onto the grid with a size corresponding to the map frame size using affine transformation. Rectification was then carried out and the image was adjusted to the reference layer using control points. Then, for all maps, the historical local reference system was transformed into the contemporary global system.

Based on these cartographical materials, vector maps were created by screen digitalization. This allowed spatial analyses to be carried out. The V_LATE add-on of the package ArcGIS® ver. 10.5.1 was used to calculate the changes in area of selected glades. The lack of accurate data from previous years meant that detailed land cover analysis was possible only for 2015. A topology construction tool was used to detect and eliminate errors generated during screen digitalization.

## 2.3. Analysis of Subjective Nature of Landscape

This analysis consists of landscape values evaluation and landscape services assessment. Landscape has a multisensory character, which is perceived through the senses—sight, hearing,

smell, touch and taste. The link between the different senses and the landscape is not universally accepted [64]. The most developed trend in landscape perception studies includes the analysis of perception and assessment of visual stimuli [65,66]. However, aesthetics cannot under any circumstances be reduced to the aesthetic values of form and the visual. Research has also been carried out concerning soundscapes [67] and smellscapes [68].

In this article, only visual stimuli are considered. One of the reasons for this is that, in the process of landscape perception, sight plays a dominant role and constitutes 85% of the total sensory perception. Secondly, the author's aim is not a detailed analysis of the stimulus aspects of pasture landscapes. For this purpose, the scenic values of the glades were analysed in accordance with the methodology proposed by Rogowski [69]. He has developed a method for multicriterial assessing the trails scenic values. One of its elements is the assessment of scenic values, which was used in this article. The criteria are presented in Table 2.

**Table 2.** The grading scale of scenic values according to panoramas from mountain meadows and glades.

| Criterion | Points |
|---|---|
| The visibility range | |
| 0–1.5 km | 1 |
| 1.5–3 km | 2 |
| 3–6 km | 3 |
| 6–12 km | 4 |
| >12 km | 5 |
| The horizontal range | |
| <60° | 1 |
| 60–120° | 2 |
| 120–240° | 3 |
| >240° | 4 |
| The vertical range | |
| <25° | 1 |
| 25–50° | 2 |
| 50–75° | 3 |
| >75° | 4 |
| Number of plans in the landscape | |
| Each plan | 1 |
| The landscape mosaicism | |
| Each land cover type | 1 |
| Dominant elements in the landscape | |
| Each dominant | 1 |

The assessment of the other values of the pasture landscapes such as antiquity, historicity, aesthetic value, authenticity, harmony, uniqueness, content, emotional and integration values, and usability was carried out on the basis of the method proposed by Myga-Piątek [70]. She claims that limitation to the determination of physicochemical characteristics and parameters of the bio- and energy cycle in the landscape does not give a complete depiction of landscape. This depiction should be complemented by value assessment. Intangible interpretations of landscapes include cognitive and perceptual aspects of the landscape, stressing that the landscape is not only a material entity of the physical world but also its representation in human mind [71]. The criteria are given in Table 3.

The assessment of landscape services was descriptive based on the classification of ecosystem services CICES [72]. A detailed analysis of this issue goes beyond the scope of the article.

**Table 3.** Criteria of landscape values assessment.

| Criterion | | Points |
|---|---|---|
| antiquity | 3 | landscapes with cultural elements older than 300 years |
| | 2 | landscapes with cultural elements aged 300–100 years |
| | 1 | landscapes with cultural elements younger than 100 years |
| historicity | 3 | landscapes in which recorded historical events have national scale and significance |
| | 2 | landscapes in which recorded historical events have regional scale and significance |
| | 1 | landscapes in which historical events were not recorded or did not occur |
| aesthetic value | 3 | landscapes with high aesthetic value |
| | 2 | landscapes with slightly disturbed aesthetics |
| | 1 | landscapes with aesthetics disturbed by the progressive transformations of the original landscape form |
| authenticity | 3 | landscapes with fully authentic structural elements |
| | 2 | landscapes with slightly transformed structural elements |
| | 1 | landscapes with structural elements completely distorted out of step with their former function and form |
| harmony | 3 | full compositional compatibility giving a sense spatial order and continuity of functions |
| | 2 | slight disturbance of spatial order and considerable continuity of functions |
| | 1 | considerable disturbance of spatial order due to lack of continuity of functions |
| uniqueness | 3 | original and unique landscapes on a national scale |
| | 2 | original and unique landscapes on a regional scale |
| | 1 | landscapes with typical and repetitive features |
| content | 3 | significant and easily determined symbolism of the landscape with distinctive "genius loci" with nationwide significance |
| | 2 | symbolism significant on a regional scale |
| | 1 | landscapes whose content is non-symbolic |
| emotional and integration value | 3 | the community shows close emotional ties with the place |
| | 2 | community connections with the place concern only selected social and age groups |
| | 1 | attachment and affiliation relationships have been broken and the local community no longer identifies with the tradition of the place and does not care about the state of the space |
| usability | 3 | landscapes used in accordance with a traditional function that brings economic benefits |
| | 2 | landscapes used in accordance with the traditional function which is uneconomic and therefore exposed to functional transformations and changes in land use patterns |
| | 1 | no use in accordance with the original function |

## 3. Results

### 3.1. State and Changes of Pasture Landscape

At present, the Carpathian pasture landscapes are preserved to varying degrees. The state of their preservation can be defined on the basis of a land use and land cover map drawn up within the historical range of the glades (Figure 2). The Austrian cadastral maps from 1848 were the basis for defining the historical range of the mountain meadows and glades.

The percentage of the various forms of land cover within the historical range of the analysed glades varies (Table 4). Within the glades where landscape conservation is carried out (mowing of vegetation and sheep grazing), non-forest vegetation communities still occupy large areas (Radziechowska glade, Bendoszka glade), whereas in the unused glades forests predominate (Jaskowa glade).

Among the non-forest vegetation on the Radziechowska glade, *Agrostis capillaris-Festuca rubra* dominates. Furthermore, the glade is inhabited by *Hieracio (vulgati)-Nardetum*, a community of *Holcus mollis* and *Juncus effusus*, and *Carici canescentis-Agrostietum caninae*. In many places, the glade is covered by *Vaccinium myrtillus*. The association of *Rubetum idaei* also occurs. Jaskowa glade, in turn, is almost entirely covered by *Vaccinium myrtillus*, whereas on Bendoszka glade *Gladiolo-Agrostietum* occurs.

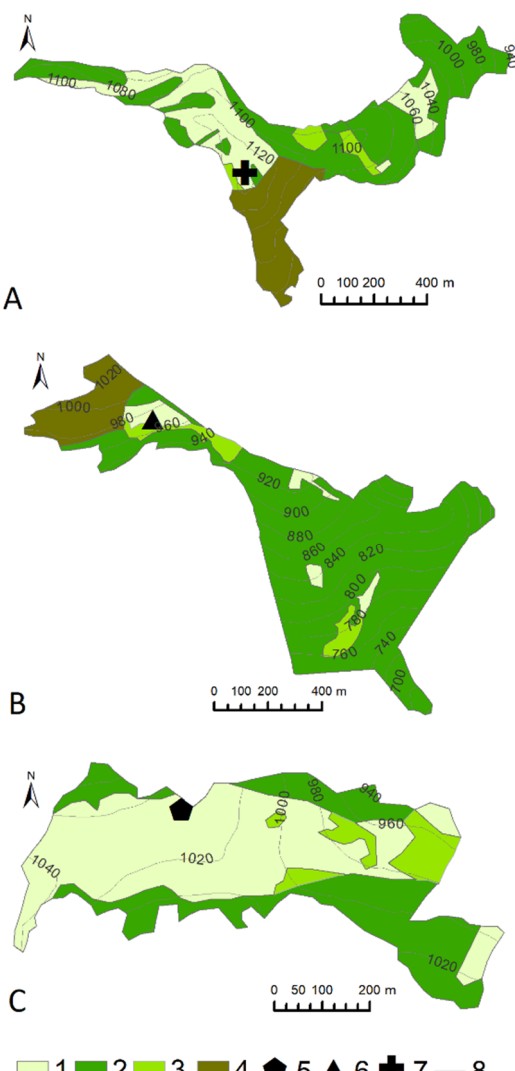

**Figure 2.** Land cover and land use of the selected glades in 2015. (**A**)—Bendoszka glade, (**B**)—Jaskowa glade, (**C**)—Radziechowska glade. Explanations: 1—non-forest communities, 2—forest communities, 3—areas of secondary succession, 4—clearcutting, 5—new pasture shelters, 6—ruined shelters, 7—other objects, 8—contour lines.

**Table 4.** Land cover of selected glades in 2015.

| Land Cover Type | Bendoszka Glade | | Jaskowa Glade | | Radziechowska Glade | |
|---|---|---|---|---|---|---|
| | [ha] | [%] | [ha] | [%] | [ha] | [%] |
| Grasslands | 12.7 | 26 | 3.2 | 5 | 14.7 | 57 |
| Areas of secondary succession | 2.1 | 4 | 3.1 | 5 | 2.1 | 8 |
| Forests | 25.1 | 52 | 49.8 | 79 | 8.9 | 35 |
| Clearcutting areas | 8.7 | 18 | 7.1 | 11 | 0.0 | 0 |
| Total | 48.6 | 100 | 63.2 | 100 | 25.7 | 100 |

The proportion of forests is high within the historical range of the glades, ranging from 35% in Radziechowska glade to 79% in Jaskowa glade.

The analysed glades are also different in terms of land use. Historical buildings, in the form of shepherd's shelters, have not been preserved in any of the glades. However, there is a new shepherd's hut in Radziechowska glade as a result of the "Owca Plus" Economic Activation and Cultural Heritage

Conservation Program. There are still ruins of a hut in Jaskowa glade, while in the Bendoszka glade, a viewing platform and the Millennium cross can be found.

Pasture landscapes have been transformed as a result of the interaction of related ecological, social and economic processes. As a result, the spatial extent of pasture landscapes has been reduced over the last 150 years. This can be analysed quantitatively by comparing data on the extent of the glades in the past (Figure 3). The rate of decrease in the area of the glades depends on the degree to which they have remained in use. The largest decrease in area was recorded on the currently disused Jaskowa glade (almost twentyfold), whereas, in the fully utilized Radziechowska and Bendoszka glades, the rate of shrinkage is less pronounced.

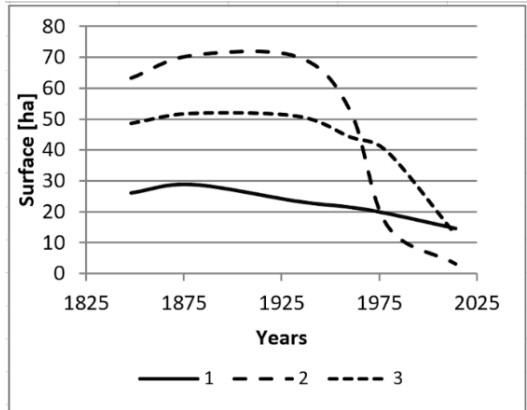

**Figure 3.** Changes in surface area of glades: 1—Radziechowska glade, 2—Bendoszka glade, 3—Jaskowa glade.

### 3.2. Pasture Landscape Values and Services

Mountain meadows and glades, as open areas, are excellent viewpoints. The scenic values of them are diverse (Table 5). This diversity is connected mainly with the horizontal range, number of plans and landscape mosaicism. The Jaskowa glade, which is disused and where forests succession is observed, received a smaller number of points. The horizontal range and number of plans is smaller than in other analysed glades. What is more, this glade has monotonous land cover as forests predominate (Table 4). In turn, there are picturesque panoramas visible from other analysed glades (Figure 4). They are characterized by vastness, multi-planarity and mosaicism.

**Table 5.** Scenic values assessment of selected glades.

| Criterion | Bendoszka Glade | Jaskowa Glade | Radziechowska Glade |
|---|---|---|---|
| The visibility range | 5 | 4 | 5 |
| The horizontal range | 4 | 2 | 3 |
| The vertical range | 4 | 4 | 4 |
| Number of plans in the landscape | 10 | 5 | 11 |
| The landscape mosaicism | 5 | 3 | 5 |
| Dominant elements in a landscape | 3 | 1 | 1 |
| Total | 31 | 19 | 29 |

It must be emphasised that glades themselves play a significant role in the attractiveness of the panoramas, which results from the seasonal variability of their colouring associated with the blossoming of vascular plants and the occurrence of wooden shepherding buildings or grazing sheep. Visual perception is complemented by smell (scent of flowers, herbs, freshly mown grass), sounds (the buzz of flying insects, crickets chirping, birds singing and sounds associated with sheep grazing), flavours (raspberries, blackberries, blueberries) and touch (a gentle breeze). These are important for increasing the attractiveness of the region for tourists, as glades help to prevent fatigue. Wandering

in uninterrupted forest becomes monotonous after some time, and the occurrence of contrasts has a stimulating effect [73].

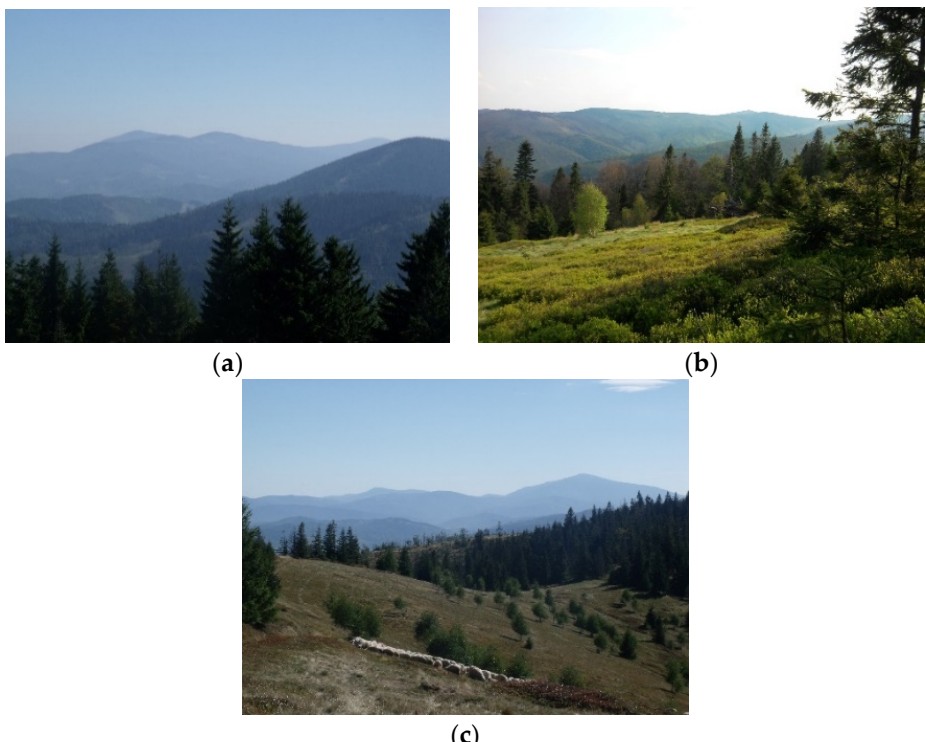

(**a**)　　　　　　　　　　　　　　　　　　(**b**)

(**c**)

**Figure 4.** Fragments of panoramas visible from (**a**) Bendoszka glade; (**b**) Jaskowa glade; (**c**) Radziechowska glade.

The analysed pasture landscapes obtained a similar sum of total points in the assessment of other values (Table 6). However, it is possible to distinguish the significance of particular values, especially when it comes to values such as historicity and authenticity. This diversity indicates the varied socio-economic significance of the glades in the past, and the varying degrees of to which they have been transformed in modern times. The largest sum of points relates to the antiquity and aesthetic value. In turn, the smallest sum of points is associated with the authenticity, harmony and usability of the Jaskowa glade. It is relevant to progressive forests succession.

**Table 6.** Pasture landscape value assessment.

| Criterion | Bendoszka Glade | Jaskowa Glade | Radziechowska Glade |
|---|---|---|---|
| Antiquity | 3 | 3 | 3 |
| Historicity | 2 | 3 | 1 |
| Aesthetic value | 3 | 3 | 3 |
| Authenticity | 2 | 1 | 3 |
| Harmony | 2 | 1 | 2 |
| Uniqueness | 2 | 2 | 2 |
| Content | 2 | 2 | 2 |
| Emotional and integration value | 2 | 2 | 2 |
| Usability | 2 | 1 | 2 |
| Total | 20 | 18 | 20 |

Based on the classification of ecosystem services CICES [72], it can be concluded that pasture landscapes can provide the following services:

- Providing food (e.g., meat, milk, cheese, blueberries, herbs) and materials (e.g., fodder).

- Regulation and maintenance, e.g., soil stabilization, water circulation regulation, microclimate regulation.
- Cultural (interaction with nature), e.g., tourism, education, shaping regional identity.

As mentioned above, a detailed quantitative analysis of this issue goes beyond the scope of this article. An overview of the methods used to assess landscape and ecosystem services can be found in the study by Gómez-Baggethun et al. [74].

## 4. Discussion

To begin with, it must be pointed out that the aim of the article is not to explain the reasons behind changes in pasture landscapes. Changes in specific landscape elements have been analysed throughout the Carpathians, in varying degrees of detail (e.g., Munteanu et al. [75], Sobala et al. [56]). However, some issues were presented in the results chapter. The aim is to indicate that mountain pasture landscapes should be included in the typologies of landscapes which were created in recent years in many countries inter alia in connection with the implementation of the European Landscape Convention. What is more, it must be emphasised that due to the multidisciplinary nature of landscape research, only some examples of numerous methods were used in this paper. However, it must be pointed out that there are a lot of different methods used in landscape research [76]. Material and intangible elements are closely interrelated and influence each other; therefore, it is argued that it is incorrect to consider them separately [77,78].

Postulating the inclusion of Carpathian pasture landscapes in national typologies of landscapes, in the first instance, reference to the definition of landscape according to the European Landscape Convention [27] should be made. By adopting a definition of landscape which takes in the whole of the national territory, the European Landscape Convention requires parties to incorporate landscape into treatment of all types of area and into all policy areas [79]. Mountain meadows and glades can be treated as an area perceived by people, whose character is the result of the action and interaction of natural and human factors. As has been shown above, mountain meadows and glades occupy a specific part of the terrain, which can be presented on a map as a set of natural and anthropogenic objects. As a result of the impact of various processes, these are being transformed, which can be demonstrated by comparing the content of maps from different time periods. The particular components of mountain meadows and glades, as well as the totality created by them, affect first of all the sense of sight (landscape physiognomy), but are also the source of other stimuli that affect other human senses. In addition, these areas are characterized by specific values and are the source of real and potential services. In this sense, the mountain meadows and glades can be treated as a separate type of landscape (Figure 5a). This is reflected widely in the literature [4,9,16]. The integrated interpretation of landscape combines these five landscape concepts, viewing landscape as a totality. Such an approach, without separating the different dimensions of the landscape, enables the management and organization of the landscape through a cultural and natural aspects as well as tangible and intangible ones. This meets the most important contributions of the European Landscape Convention that landscape planning, management and protection issues must be seen holistically and should be coordinated [64].

The possibility of defining mountain meadows and glades through the prism of landscape definition is not yet sufficient reason to include them in national typologies of landscapes. On the other hand, some nationwide landscape typologies cited in the introduction assume that the mountain meadows and glades are a component of forest landscapes (Figure 5b). In this sense, they form patches of land cover, distinguishing themselves from the forest background and forming landmarks in the Carpathian forest landscapes [54]. They constitute a material record of the management of mountain areas by humans for the needs of the pastoral economy [57].

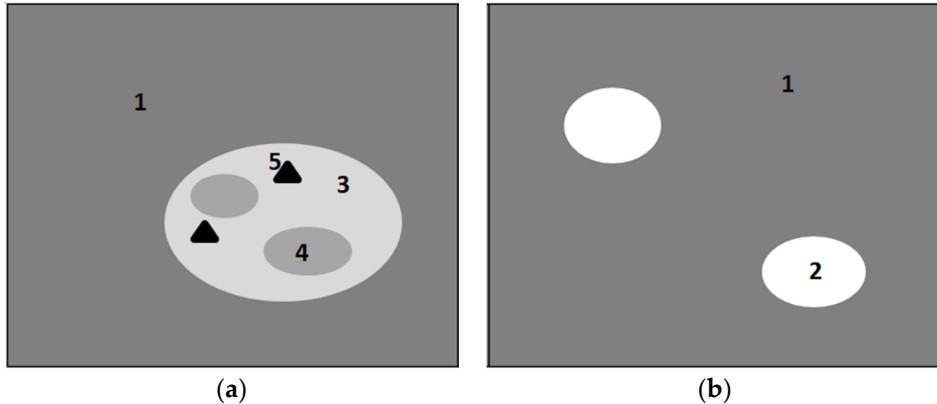

**Figure 5.** Different ways of including mountain meadows and glades in landscape typologies: (**a**) Pasture landscapes; (**b**) Mountain meadows and glades as an element of forest landscape. Explanation: 1—forest landscape, 2—glade as an element of forests landscape, 3—pasture landscape; 4, 5—pasture landscape elements.

Analysis of existing methods of landscape delimitation and classification (i.e., Majchrowska [80]; Mücher et al. [40]; Bezek et al. [81]) indicates that the typology of current landscapes should:

- Refer to landscapes distinguished as spatial units with specific boundaries and sizes.
- Refer to contemporary landscapes.
- Take into account directly the diversity, surface share and spatial layout (texture and composition) of real objects in space.

Including mountain pasture landscapes in typologies developed by individual countries for the landscape protection and management purposes (i.e., implementing the European Landscape Convention) should be consistent with these demands.

The size of the distinguished landscape units seems to be an unresolved issue. The size of relatively homogeneous land cover forms is very diverse. While pasture landscapes still occupy significant areas in the countries of tropical and subtropical climate zones, in the More Economically Developed Countries of the temperate zone, their range during the 20th century has significantly shrunk and, contemporarily, mountain pasture landscapes are usually small enclaves [82]. Sometimes regional landscapes are too small to be located on a map of Europe [39]. A minimal area of distinguished landscape units on Pan-European landscape maps are too small to indicate mountain pasture landscapes. For instance, in the European Landscape Classification (LANMAP) polygons smaller than 11 km$^2$ were removed and integrated with the adjacent polygon. It was due to the fact that general assumption was that the LANMAP would present a stronger generalisation and simplification than the national classifications. Nevertheless, this was not the case for Switzerland, Germany, the Netherlands, Spain and Norway [83]. The LANMAP gives a consistent view across Europe and provides a common language and classification system, but cannot replace any of the national landscape classifications [40]. Similarly, landscape units distinguished in the typology of European agricultural landscape were characterized at a 1 km$^2$ resolution based on Europe-wide datasets that represent land cover, landscape structure and land management intensity. It was necessary to reduce the complexity in agricultural landscapes to manageable units that could be an interesting target for policy-making at the European scale [41]. In turn, in the nationwide typologies of landscapes, the authors do not indicate a minimal area of distinguished landscape units. Their size should depend on purpose, area, scale and method of landscape identification. Other units, both in terms of size and distinguishing criteria, will be needed for studies on a national scale (assessment of landscapes throughout the country), others on a regional scale, and still others on a local scale (assessment of landscapes of a commune, national park, etc.) [84]. The fact that mountain pasture landscapes are not distinguished as a separate type/subtype of landscape in nationwide typologies can be explained by their intended use for small-scale studies of the entire

country. On the other hand, some of the landscape units separated on the basis of national typologies occupy smaller areas than the area of some of the Carpathian forests and glades [45–47]. According to authors of the European Landscape Classification, improvements are needed in terms of the spatial identification of certain landscape types, e.g., coastal dunes. The LANMAP still lacks much information at the regional level about cultural–historical and socio-economic aspects that are crucial for many regional applications [40].

Another issue concerns the current state of landscapes. Pasture landscapes in the Carpathians, like in other mountains in Europe [85], are contemporary landscapes, although often the factors that contributed to their formation currently do not affect them or have been modified. This is a threat to their survival [57]. Beginning in the second half of the 19th century, the gradual fall of the pastoral economy in the Western Carpathians contributed to the secondary succession of forest to disused mountain meadows and glades (Figure 2, Figure 3). This resulted in a significant increase in forest areas and, as a result, the closing of the landscape [75,86] that is typical of many mountain areas [87]. This process was accompanied by the disappearance of traditional buildings [57]. The 'ancient' forms of traditional hay-making structures are becoming a relic all over Europe. Hay-making structures have been mostly preserved in connection with traditional agricultural landscapes, and particularly in the more remote regions or where associated with strong cultural identity [88]. As a result of evolutionary changes of the pasture landscapes into forest or settlement landscapes, the Carpathian pasture landscapes in some areas are considered by Antrop [89] to be relict or fossil landscapes. Although in recent years, sheep grazing has been restored in many glades, its organization differs from the traditional methods, which is related to socio-economic determinants which are different from those in the past [5]. In these areas, pasture landscapes have also undergone evolutionary transformation, but are considered by Antrop [89] to be permanent landscapes (continuing landscapes), the maintenance of which is not threatened under the condition of active conservation. For example, grassland patches enclosed in a forest matrix disappearing progressively in the Italian Alps, are today a matter of high concern from conservation point of view [90,91]. Nevertheless, Slámová et al. [92] pointed out that historical rural landscape conservation practice is still very poor. Although mountain meadows and glades are of minor economic importance in many regions nowadays, they still have a high significance for people and in some countries are even part of the national heritage. Considering the disappearance and ongoing abandonment of historical rural landscapes in Europe, the preservation of these landscapes is an issue of growing importance [88]

As was indicated by Solon et al. [41], it is necessary to precisely identify and assess current landscapes for effective landscape conservation. A good understanding of landscapes is essential for their assessment, protection, management and planning [93]. As there are many regional differences in landscape features, it is essential to strike the right balance between reducing the inherent complexity and maintaining an adequate level of detail [40,94]. Naranjo [93] claims that the delineation of landscape typologies on national and regional scales must be based on the principal arrangements of a territory's structural features and the main land uses, taking into account its cultural traditions and history. It seems that the proposed nationalwide typologies do not fully utilise the scope of systematization, which is useful, or even indispensable, for the categorisation and valorisation of the landscape for the needs of effective and sustainable landscape policy, especially at the local level. Chmielewski et al. [43] postulate the distinction of the next hierarchical level (known as landscape form), taking into account local landscape features. On the one hand, this will give rise to the need to undertake research on the identification of landscapes with very characteristic local features. These actions, in turn, can be important for the effective protection of unique landscape features. On the other hand, the inclusion of mountain pasture landscapes into the existing typologies, taking into account the existing hierarchy of division, may constitute a stronger basis for the protection of pasture landscapes due to the national significance of its typology [50].

## 5. Conclusions

Mountain pasture landscapes are real landscapes created by a set of specific natural and anthropogenic objects constantly subjected to transformation, which are a source of incentives, values and real and potential services.

In the Pan-European and national landscape typologies developed in individual European countries for the purposes of the implementation of the European Landscape Convention, the mountain pasture landscape type has not been distinguished. Distinguishing this type of landscape could be important for the effectiveness of conservation and landscape management activities, as these activities depend on the detailed identification and assessment of landscapes. An alternative solution may be to distinguish the next hierarchical level within existing typologies, taking into account local landscape features. National typologies of landscapes are a tool for organizing knowledge about the levels and forms of contemporary landscape transformations and mapping this diversity. In the face of significant transformations of pasture landscapes in the Carpathians, including them in national typologies seems to be an important issue for ensuring their survival.

**Funding:** This research received no external funding.

**Acknowledgments:** I gratefully acknowledge three anonymous reviewers for their constructive comments.

**Conflicts of Interest:** The author declares no conflict of interest.

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
