# Peer review of "Mountain Meadows and Glades of the Carpathians—Type or Element of Landscape? The Problem of Delimitation and Typology of Mountain Pasture Landscapes"

_sustainability, doi:10.3390/su12093707_

Round 1
Reviewer 1 Report
The author postulates to include pasture landscapes of the Carpathians into national landscape typologies that are being developed in landscape identification and classification studies under the states that are parties to the European Landscape Convention. Generally, this is an apt postulate, because pastoral landscapes have valuable biodiversity and heritage values as a reminder of an interesting type of agricultural culture. Thus, their presence in landscape typologies could raise their rank and make problems with their current management more visible.
To support the postulate, the author conducts several analyses, which are to show that mountain meadows and glades possess characteristics of a separate type of landscape, taking as the example three pastoral areas from the western part of the Carpathians. The author analyzed the material and immaterial features of pastoral areas, relating the analyzes to the definitions of landscape, among others to the definition given by the European Landscape Convention, but also to others definitions which can be found in the scientific literature. The analyses of diverse properties of selected fragments of mountain meadows and glades are to prove that they fulfil the definition of landscape.
In Europe we already have examples of typologies that include pastures and meadows: in Austria, Belgium, there is the typology of European agricultural landscapes (the Eucaland project), as well as a study on the typology of Polish cultural landscapes, so please give references. If there are no Carpathian pastures in the Polish study, then it is a matter of scale. The problem of the scale was noticed by the author and his solution is to create a hierarchical typology in which pastures would be included. So is the postulate of separating pastoral landscapes only applicable to the Carpathian region?
I see no strong connection between the analysis of material and non-material aspects of the mountain meadows and glades of the Carpathians and the postulate of separating pastoral landscapes. The analysis is rather a comparison of the three test areas at different angles, rather than proving that meadows and glades have the same features constituting a separate type of landscape as other types of landscape. I propose to include more analyses related to objective (material) features other than area changes. At least, please suggest the type of analysis or features that should be analyzed.
Some specific comments:
Line 133 mean annual temperature
Line 138 ridges include
Line 152 similar scales - the scales of maps used in the analysis are not similar 1: 2,880 and 1:100,000
Line 227 lacking caption of the table (no4)
Line 257 lacking caption of the table (no5)
Reviewer 2 Report
Dear Author,
you have done a good research on state and changes of pasture landscape in selected study area as well as on pasture landscape values and services. Taking into account your article title "Mountain meadows and glades of Carpathians - type or element of landscape? The problem of delimitation and typology of pasture landscapes" the text of the article somehow does not clearly follow this title. In discussion part you articulate a need for incorporating the meadows as a part of landscape typology but previous article parts sleak about values and landscape chenges over a certain period. The article is not coherent at all and seems to me a very chaotic.
Specific comments:
Abstract:
1.Please state clearly what is the aim of your contribution (once you define this article (reasearch) objective you sholud continue in a way that support and make argumentation of your research properly and coherently).
2. line 9 - is this statement supported by a serious research of countries - signatories of Landscape Convention? It is a very serious statement. In several countries, for example, any official landscape typology do not exist. If you want keep this sentence you should have some proof for that. It seems that you have data mostly from Poland Slovakia. What about other countries? Maybe it would be helpful to limit it to Central European countries or so.....
3.line 12 -" artticle demonstrates that pasture landscapes should be......" From my oint of view, the article demonstrates values and changes of meadows and glades without any demonstration of the need to be included in national landscape typologies.
Introduction:
Again, there a need to include several lines on the landscape typologies in Europe or just in Central Europe supported by relevant data sources (references)
Materials and methods:
Please create a new subtitle - 2.4. Analysis of landscape typology in Europe
Results
Please reate a new subtitle - The present situation on meadows as a part of landscape typology
Conclusions
If you make changes in previous article parts, then please make a relevant reflection in conclusions.
References:
I would to recommend to include or substitute other references by the following ones:
1.ŠPULEROVÁ, Jana** – KRUSE, Alexandra – BRANDUINI, Paola – CENTERI, Csaba – EITER, Sebastian – FERRARIO, Viviana – GAILLARD, Bénédicte – GUSMEROLI, Fausto – JURGENS, Suzan – KLADNIK, Drago – RENES, Hans – ROTH, Michael – SALA, Giovanni – SICKEL, Hanne – SIGURA, Maurizia – ŠTEFUNKOVÁ, Dagmar – STENSGAARD, Kari – STRASSER, Peter – IVASCU, Cosmin Marius – ÖLLERER, Kinga. Past, present and future of hay-making structures in Europe. In Sustainability [serial], 2019, vol. 11, no. 20, article no. 5 581.
2.Slámová, M.; Belcáková, I. The Role of Small Farm Activities for the Sustainable Management of Agricultural
Landscapes: Case Studies from Europe. Sustainability 2019, 11, 5966.
They are much more suitable. Furthermore they can provide you a lot of useful information on the topic mapping the European context
Reviewer 3 Report
This research work presents a current topic of great interest for the administration of cultural heritage and world cultural landscapes. Emphasizing the absence of a landscape typology in a specific Polish geographic area: mountain meadows and clearings in the Carpathians.
It has good methodological support and support in the cartographic materials used for the analyzes.
The vision of the article is somewhat limited, since if its main objective is the inclusion of pasture landscapes in the type or element of the Polish landscape and, although it is largely based in the text of the European Landscape Convention of 2000, does not take into account in its defense, other fundamental aspects established in the same document (CEP), launched by the Council of Europe, such as: promoting the protection, management and organization of the landscape through a cultural, heritage and natural vision , through coordination between European policies on cultural heritage, the environment and spatial planning (there are no bibliographic references in this regard).
In this framework, the complete defense claimed by the author of pasture landscapes will have a place.
To support your defense objective, I recommend reading Prieur, M. and Durousseau, S. 2006. Landscape and sustainable development ...
As well as the text of the 15th Meeting of the Council of Europe of the workshops for the implementation of the Europan Landscape Convention. Sustainable landscapes and economy: on the invaluable natural and human value of the landscape.
Round 2
Reviewer 2 Report
I can accept the updated manuscript´s version
Author Response
I would like to thank you for all comments that enable to improve my paper.